# Crystal structures of DNA polymerase I capture novel intermediates in the DNA synthesis pathway

Nicholas Chim[1†], Lynnette N Jackson[1†], Anh M Trinh[1], John C Chaput[1,2,3]*

[1]Departments of Pharmaceutical Sciences, University of California, Irvine, California; [2]Department of Chemistry, University of California, Irvine, California; [3]Department of Molecular Biology and Biochemistry, University of California, Irvine, California

**Abstract** High resolution crystal structures of DNA polymerase intermediates are needed to study the mechanism of DNA synthesis in cells. Here we report five crystal structures of DNA polymerase I that capture new conformations for the polymerase translocation and nucleotide pre-insertion steps in the DNA synthesis pathway. We suggest that these new structures, along with previously solved structures, highlight the dynamic nature of the finger subdomain in the enzyme active site.

DOI: https://doi.org/10.7554/eLife.40444.001

*For correspondence:
jchaput@uci.edu

†These authors contributed equally to this work

Competing interests: The authors declare that no competing interests exist.

## Introduction

DNA polymerase I (DNAP-I) has long been viewed as the canonical model for DNA synthesis in cells (*Lehman et al., 1958*). Structural insights into the mechanism of DNA synthesis have been obtained from crystal structures of a thermostable bacterial (*Geobacillus stearothermophilus, Bst*) DNAP-I large fragment that retains catalytic activity inside the crystal lattice (*Johnson et al., 2003*; *Kiefer et al., 1998*). The prevailing mechanism invokes the use of a distinct pre-insertion site, observed in the translocated product of *in crystallo* catalyzed primer-extension reactions where dNTP substrates are soaked into pre-formed crystals of DNAP-I bound to a primer-template duplex (*Figure 1—figure supplement 1*)(*Johnson et al., 2003*; *Kiefer et al., 1998*). The pre-insertion site is a hydrophobic pocket located between the O and O1 helices of the finger subdomain where the n + 1 templating base resides prior to forming the nascent base pair with the incoming dNTP substrate (*Johnson et al., 2003*). However, the pre-insertion site has not been witnessed in polymerases with homologous active sites (*Eom et al., 1996*; *Li et al., 1998*; *Yin and Steitz, 2002*), implying that DNAP-I follows a complex enzymatic pathway that contains numerous intermediates, many of which have not yet been observed in protein crystals. Here we report five crystal structures of DNAP-I that capture new conformations for the polymerase translocation and nucleotide pre-insertion steps in the DNA synthesis pathway. Together, these structures provide new insight into the mechanism of DNA synthesis and highlight the dynamic nature of the finger subdomain in the enzyme active site.

## Results and discussion

Recognizing that *in crystallo* and solution catalyzed enzymatic reactions can produce different structural results with potentially different functional interpretations (*Ehrmann et al., 2017*), we chose to investigate the translocated intermediates of DNAP-I using a direct crystallization method that involves solving crystal structures of the enzyme-product complex obtained from primer-extension reactions performed in solution rather than inside the environment of a protein crystal. In these

**eLife digest** DNA molecules consist of two separate strands that spiral around each other to form a structure called the double helix. Each strand contains repeating units, with every unit consisting of a phosphate group and a sugar molecule bound to one of four bases. The two strands are held together by bonds between the bases.

When a cell divides, it needs to make a copy of the DNA, so that each new cell will have an exact replica from the old cell. During this process, the helix unwinds and enzymes called polymerases produce new strands (using the old ones as a template). Each strand is copied by adding new bases one at a time. Every time a new base is added, the polymerases must modify their structures several times. If this process becomes faulty, it can lead to various diseases, including cancer.

Scientist often use a technique called X-ray crystallography to study intermediate structures of frozen polymerase crystals as the enzyme constructs DNA. Yet, to fully understand the mechanisms of DNA synthesis all intermediate structures need to be identified.

Now, Chim, Jackson et al. used a particular method for making frozen polymerase crytals by allowing the enzyme to add new bases in liquid form. The reaction was then frozen and X-ray crystallography was used to take images. This modified method captured different steps in the process and detailed how the enzyme adjusts its structure as it moves along the template strand.

The intermediate structures that Chim, Jackson et al. uncovered may help scientists develop new biotechnologies and medicines. Understanding how polymerases modify their form while making DNA copies could lead to better therapies for diseases in which this process has become faulty, like cancer.

DOI: https://doi.org/10.7554/eLife.40444.002

reactions, the starting enzyme-primer-template complex was incubated with solutions of either buffer, dTTP, or dTTP and dATP for 30 min at 37°C. Following primer-extension, the enzyme-product complex was crystallized and cocrystal structures of Bst DNAP-I were solved to resolutions of 1.5 – 2.0 Å (*Table 1*). This approach was used to obtain high resolution structures of DNAP-I for the starting primer-template complex (n) and two translocated products obtained for the n + 1 and n + 2 nucleotide addition steps using the same primer-template duplex (n) described in previous *in crystallo* studies (*Figure 1a*)(*Johnson et al., 2003*).

Structures of the enzyme-primer-template complex (n) before catalysis reflect the initiation step of DNA synthesis. Superposition of the new structure obtained for the initiation step against the previously solved structure reveals that both structures adopt the same active site conformation (*Figure 1—figure supplement 2a*). This result implies that any structural differences observed between the translocated product of solution and *in crystallo* catalyzed reactions should be due to the catalysis environment rather than the starting polymerase conformation.

To evaluate the elongation step of DNA synthesis, the translocated products obtained from solution and *in crystallo* catalyzed primer-extension reactions were compared, both globally and locally within the enzyme active site (*Johnson et al., 2003*; *Kiefer et al., 1998*). All of the structures adopt the same overall topology commonly observed for A-family DNA polymerases (*Figure 1b*). However, careful analysis of the enzyme active site did reveal clear conformational differences between structures obtained from solution-catalyzed reactions versus those obtained from *in crystallo* catalyzed reactions (*Figure 1c,d*). The *in crystallo* catalyzed reactions adopt an active site conformation that is nearly identical to the starting conformation, which represents the initiation step of DNA synthesis (*Figure 1—figure supplement 2a*). However, the solution catalyzed reactions produce a different active site conformation that binds the duplex in a different position and base pair geometry (*Figure 1—figure supplement 2b,c*).

Major structural differences are depicted in the 2D interaction maps, which show that the solution catalyzed reactions produce a translocated product with markedly fewer contacts to the phosphodiester linkage, sugar, and nucleobase moieties of the primer-template duplex as compared to the translocated product obtained by *in crystallo* catalysis (*Figure 1—figure supplements 3* and *4*, *Supplementary file 1a*). A particularly striking example of conformational disparity is Tyr$^{714}$, a critical active site residue involved in the mechanism of DNA synthesis (*Bell et al., 1997*; *Carroll et al.,*

**Table 1.** Data collection and refinement statistics

| | N | n + 1 | n + 1, dATP soak | n + 1, dAMPNPP soak | n + 2 |
|---|---|---|---|---|---|
| Data Collection | | | | | |
| Space group | $P2_12_12_1$ | $P2_12_12_1$ | $P2_12_12_1$ | $P2_12_12_1$ | $P2_12_12_1$ |
| Cell Dimensions | | | | | |
| a, b, c (Å) | 86.1, 93.4, 105.6 | 88.1, 93.7, 105.8 | 87.1, 93.5, 105.3 | 87.44, 93.39, 104.95 | 87.0, 93.0, 104.7 |
| α, β, γ (°) | 90.0, 90.0, 90.0 | 90.0, 90.0, 90.0 | 90.0, 90.0, 90.0 | 90.0, 90.0, 90.0 | 90.0, 90.0, 90.0 |
| Resolution (Å) | 54.31–1.58 (1.64–1.58) | 46.09–1.98 (2.05–1.98) | 46.7–1.99 (2.06–1.99) | 43.72–1.74 (1.78–1.74) | 41.04–1.99 (2.06–1.99) |
| $R_{merge}$ | 0.7309 (1.35) | 0.7085 (1.389) | 0.1329 (0.7194) | 0.0567 (0.241) | 0.3279 (1.745) |
| CC1/2 | 0.842 (0.795) | 0.759 (0.543) | 0.993 (0.796) | 0.999 (0.977) | 0.991 (0.586) |
| I / $\sigma$I | 71.43 (3.56) | 43.97 (2.64) | 17.03 (2.90) | 22.78 (9.66) | 9.75 (2.49) |
| Completeness (%) | 99.98 (99.98) | 96.97 (99.15) | 99.92 (99.90) | 98.25 (99.33) | 99.90 (99.93) |
| Redundancy | 31.3 (25.0) | 12.9 (11.0) | 6.8 (4.7) | 7.0 (6.8) | 7.2 (7.3) |
| | | | | | |
| Refinement | | | | | |
| Resolution (Å) | 54.31–1.58 (1.64–1.58) | 46.09–1.98 (2.05–1.98) | 46.7–1.99 (2.06–1.99) | 43.72–1.98 (2.05–1.98) | 41.04–1.99 (2.06–1.99) |
| No. reflections | 115039 (11360) | 59886 (6056) | 59677 (5857) | 59416 (5901) | 58990 (5831) |
| $R_{work}/R_{free}$ | 0.165/0.189 (0.199/0.248) | 0.202/0.255 (0.264/0.340) | 0.184/0.219 (0.239/0.293) | 0.222/0.271 (0.225/0.281) | 0.192/0.228 (0.332/0.391) |
| No. atoms | 5961 | 4627 | 5412 | 5468 | 5453 |
| Protein | 4636 | 4627 | 4639 | 4661 | 4590 |
| Duplex | 490 | 469 | 487 | 429 | 475 |
| Solvent | 835 | 546 | 286 | 378 | 388 |
| B-factors | 26.73 | 42.07 | 45.96 | 42.25 | 39.05 |
| Protein | 25.11 | 42.17 | 45.39 | 41.74 | 38.85 |
| Duplex/dAMPNPP | 40.11 | 55.17 | 117.16 | 101.56/106.4 | 60.07 |
| Solvent | 36.64 | 40.95 | 46.01 | 41.13 | 41.37 |
| R.m.s deviations | | | | | |
| Bond lengths (Å) | 0.006 | 0.007 | 0.008 | 0.008 | 0.007 |
| Bond angles (°) | 0.82 | 0.89 | 0.84 | 1.15 | 0.85 |

*Values in parentheses are for the highest-resolution shell.

DOI: https://doi.org/10.7554/eLife.40444.009

1991). In the solution catalyzed structures, Tyr[714] stabilizes the newly formed base pair by stacking above the primer strand, while this residue stacks above the template strand in the *in crystallo* catalyzed structures (*Figure 1c,d*). Importantly, the pre-insertion site is not observed in the solution catalyzed reactions due to a kink in the O-helix, which abrogates the O-O1 loop in the finger subdomain (*Figure 1d*). Absent a hydrophobic pocket, the n + 1 nucleotide in the template strand stacks against Tyr[719] in the O1 helix, which positions the base for a subsequent round of catalysis. The solution catalyzed structures obtained for the n + 1 and n + 2 translocated products adopt identical active site conformations (*Figure 1 – figure supplement 2d*), which together represent a new intermediate along the DNA replication pathway of Bst DNAP-I.

Next, we examined whether a solution catalyzed conformation could be converted to an *in crystallo* conformation through a round of *in crystallo* catalysis. Accordingly, dATP was soaked into a crystal of the n + 1 translocated product obtained by crystallization of a solution catalyzed reaction. Following one cycle of *in crystallo* catalysis, an n + 2 translocated structure was produced that now contained the pre-insertion site and matched the active site conformation of previous *in crystallo* results (*Figure 1 – figure supplement 2e, f*). This observation demonstrates that *in crystallo* catalysis favors an active site conformation that contains the pre-insertion site, as the same active site conformation is obtained from two different starting points.

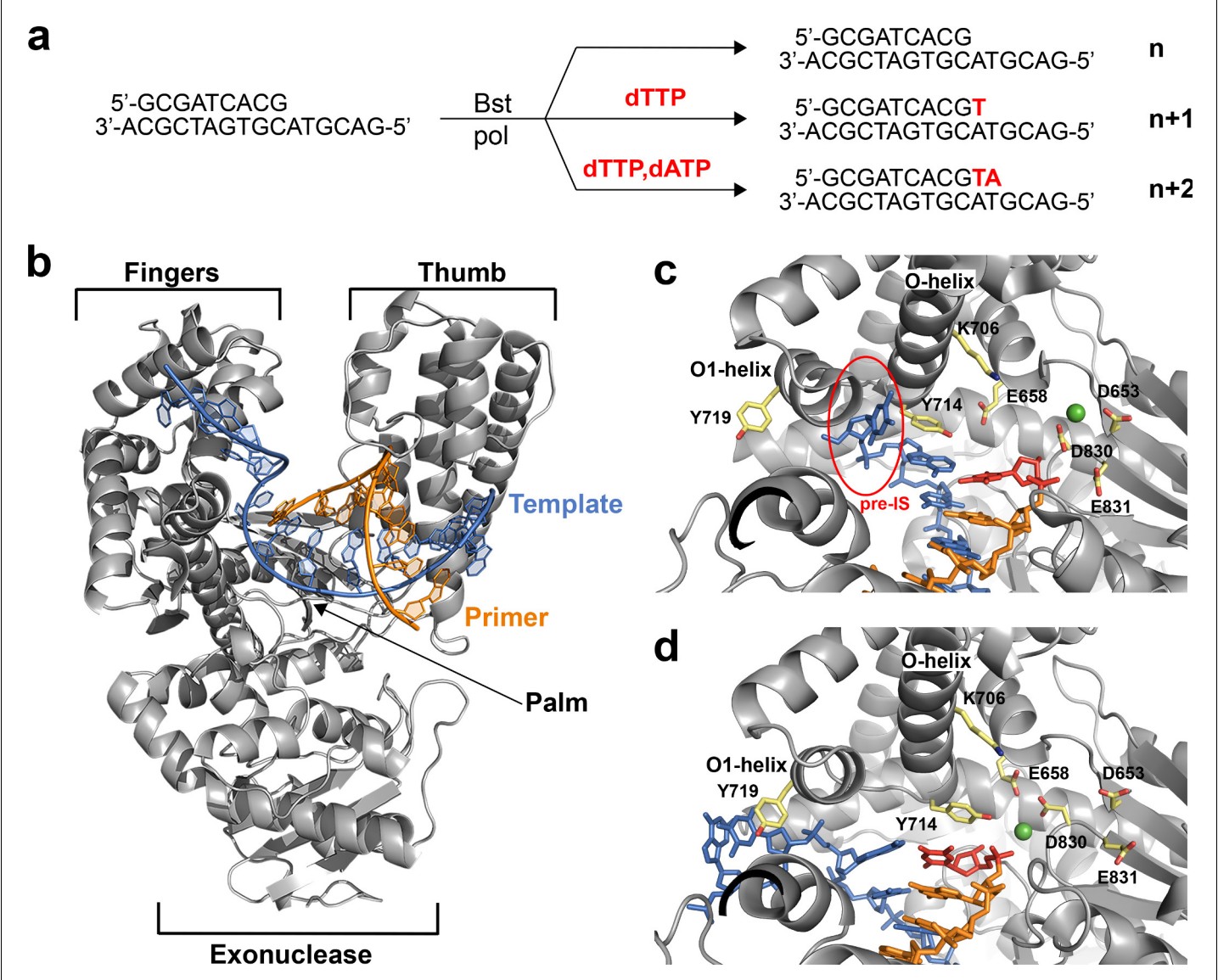

**Figure 1.** The translocation complex of Bst DNAP-I. (a) Schematic illustration of the primer-extension reactions used to generate enzyme complexes for the starting duplex (n) and translocated products of the n + 1 and n + 2 nucleotide addition steps. (b) Global architecture of Bst DNAP-I bound to the primer-template duplex (n, 6DSW). (c) The active site region of a known n + 1 *in crystallo* catalysis structure (1L3T). The pre-insertion site (pre-IS) is circled in red. (d) The active site region of the n + 1 solution-catalyzed reaction (6DSY). Color scheme: polymerase (grey), template (blue), primer (orange), magnesium ion (green), n + 1 nucleotide adduct (red), and amino acid side chains (color by atom).
DOI: https://doi.org/10.7554/eLife.40444.003

The following figure supplements are available for figure 1:

**Figure supplement 1.** Prevailing mechanism of DNA synthesis by DNA polymerase I.
DOI: https://doi.org/10.7554/eLife.40444.004

**Figure supplement 2.** Conformational changes at the active site of Bst DNAP-I.
DOI: https://doi.org/10.7554/eLife.40444.005

**Figure supplement 3.** Two-dimensional interaction map for the n + 1 structure obtained from *in crystallo* catalysis.
DOI: https://doi.org/10.7554/eLife.40444.006

**Figure supplement 4.** Two-dimensional interaction map for n + 1 structure obtained from a solution-catalyzed reaction.
DOI: https://doi.org/10.7554/eLife.40444.007

**Figure supplement 5.** Crystal structures obtained from solution catalyzed reactions resemble active site conformations observed in 'distorted' conformations.
DOI: https://doi.org/10.7554/eLife.40444.008

Interestingly, the translocated product obtained from the set of solution catalyzed reactions is similar to known Bst DNAP-I structures solved with duplexes that contain damaged DNA intermediates and active site mutations (*Figure 1—figure supplement 5*, *Supplementary file 1b*). These structures were previously thought to contain a distorted active site conformation due to the position of Tyr[714] relative to its conformation in the *in crystallo* catalysis structures (*Gehrke et al., 2013*; *Johnson and Beese, 2004*; *Wang et al., 2012*). However, given the homology of these structures to the translocated product of solution catalyzed reactions, we postulate that Tyr[714] functions as a regulatory checkpoint in the mechanism of DNA synthesis by evaluating the geometry of the newly formed base pair.

Next, we wondered whether the mechanism of DNAP-I included the formation of a pre-insertion complex, which is a ternary structure different from the previously discussed pre-insertion site observed in the binary structure of *in crystallo* catalyzed primer-extension reactions. Previously, Wu and colleagues solved the ternary structure of a mutant version of Bst DNAP-I bound to an incoming dATP substrate (*Miller et al., 2015*). Although that structure was originally described as an open ternary complex, presumably to avoid confusion with the pre-insertion site, it resembles the pre-insertion complex first observed in Klentaq1 (*Li et al., 1998*). The key difference between the open ternary and pre-insertion complex is whether the incoming nucleotide is paired opposite the templating base or an active site residue (*Doublié et al., 1998*; *Yin and Steitz, 2004*). Since the structure by Wu and colleagues shows the incoming substrate paired opposite Tyr[714], it should be considered a pre-insertion complex.

We demonstrated that the wild-type polymerase is also capable of forming a pre-insertion complex by solving the ternary structure of the enzyme bound to the non-hydrolyzable analog, dAMPNPP. The resulting structure (*Figure 2*) closely resembles the mutant Bst polymerase structure determined by Wu and colleagues and shows Tyr[714] paired opposite the incoming nucleotide (*Miller et al., 2015*). Although the phosphate tail shows nearly 100% occupancy, the sugar and nucleobase moieties are flexible, which is consistent with the dynamic properties of the incoming nucleotide in an open polymerase conformation. Nevertheless, the structure shows that the incoming nucleotide is stabilized by polar contacts to the negatively charged triphosphate moiety. These

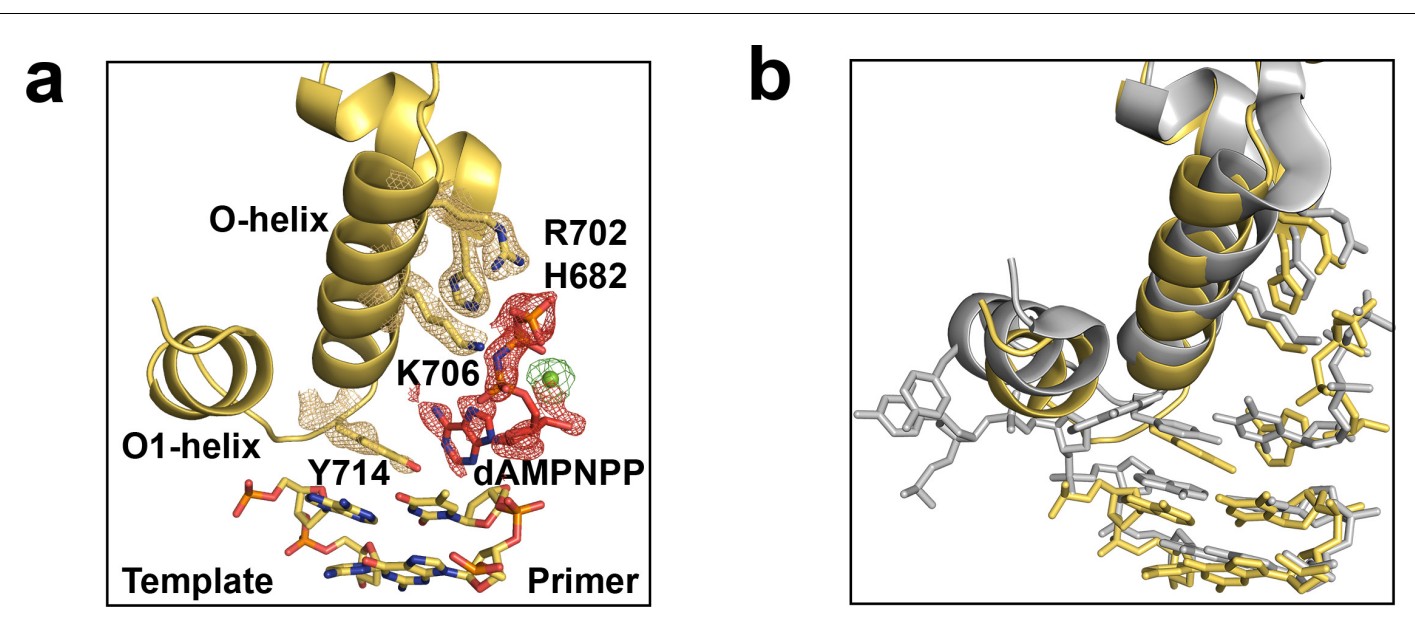

**Figure 2.** The pre-insertion complex of Bst DNAP-I. (a) An open ternary structure of Bst DNAP-I (yellow) with a primer-template duplex (color by atom), non-hydrolyzable dATP analog (dAMPNPP, red), and magnesium ion (green) bound in the enzyme active site. Superimposed on the stick model is a 2Fo-Fc omit map contoured at 2.0σ for interacting residues, yellow mesh, and Fo-Fc omit maps contoured at 1.0σ for dAMPNPP and magnesium, red and green mesh, respectively. (b) Comparison of the new ternary structure (yellow, 6DSU) superimposed on a mutant Bst DNAP-I structure solved with dATP bound in the enzyme active site (grey, 4YFU).
DOI: https://doi.org/10.7554/eLife.40444.010

observations demonstrate that Bst DNAP-I adopts a pre-insertion complex similar to other A-family DNA polymerases (*Rothwell and Waksman, 2005*), which clarifies an important step in the mechanism of DNA synthesis.

Based on the structures reported here, we propose a revised mechanism for DNA synthesis by DNA polymerase I. The catalytic cycle consists of four key steps that derive from high resolution structures of Bst DNAP-I and its homolog T7 RNA polymerase (*Figure 3*). Starting from the newly determined post-translocation complex, the polymerase undergoes a conformation change to adopt the pre-insertion complex with an incoming nucleotide paired opposite $Tyr^{714}$ in the enzyme active site. This conformational change involves release of the n + 1 templating base from its stacking interaction with $Tyr^{719}$ in the O1 helix and the repositioning of $Tyr^{714}$ in the enzyme active site. The enzyme then undergoes a more significant conformational change to adopt the closed ternary complex (*Johnson et al., 2003*), which defines the pre-catalytic state of the enzyme. Immediately following phosphodiester bond formation, the enzyme adopts a post-catalytic complex in which the primer has been extended by one nucleotide (*Yin and Steitz, 2004*). The enzyme then translocates to the next position on the template to initiate another cycle of nucleotide addition.

In summary, we present crystal structures of DNA polymerase I that capture the translocation and nucleotide pre-insertion steps in the DNA synthesis pathway. We suggest that these new structures, along with previously solved structures obtained by *in crystallo* catalysis, highlight the dynamic nature of the finger subdomain in the enzyme active site. Together, the new and existing structures expand our understanding of the mechanism of DNA synthesis by capturing important intermediates in a complicated reaction pathway.

# Materials and methods

## Key resources table

| Reagent type (species) or resource | Designation | Source or reference | Identifiers | Additional information |
|---|---|---|---|---|
| Strain, strain background (*E. coli*) | DH5-αderivative | NEB | C2987H | Chemically competent cells for recombinant expression of Bst DNAP-I |
| Recombinant DNA reagent | pDEST007-Bst | PMID: 20813757 | | Original expression plasmid for Bst DNAP-I |
| Recombinant DNA reagent | pGDR11 | PMID: 9401025 | | Expression plasmid for Bst DNAP-I used in this study |
| Sequence-based reagent | Bst_for | IDT | | 5'-ATC*CATATG*GCATTT ACGCTTGCTGAC-3' |
| Sequence-based reagent | Bst_rev | IDT | | 5'-ATGCGGC*GGTCTC*C TCGAGTCATTATTT CGCATCATACCACG-3' |
| Sequence-based reagent | DNA template | IDT | | 5'-GACGTACG TGATCGCA-3' |
| Sequence-based reagent | DNA primer | IDT | | 5'- GCGATCACGT-3' |
| Software, algorithm | XDS | PMID: 20124692 | RRID: SCR_015652 | |
| Software, algorithm | Phaser | PMID: 19461840 | RRID: SCR_014219 | |
| Software, algorithm | Phenix refine | PMID: 22505256 | RRID: SCR_014224 | |
| Software, algorithm | Coot | PMID: 20383002 | RRID: SCR_014222 | |
| Software, algorithm | Molprobity | PMID: 2057044 | RRID: SCR_014226 | |

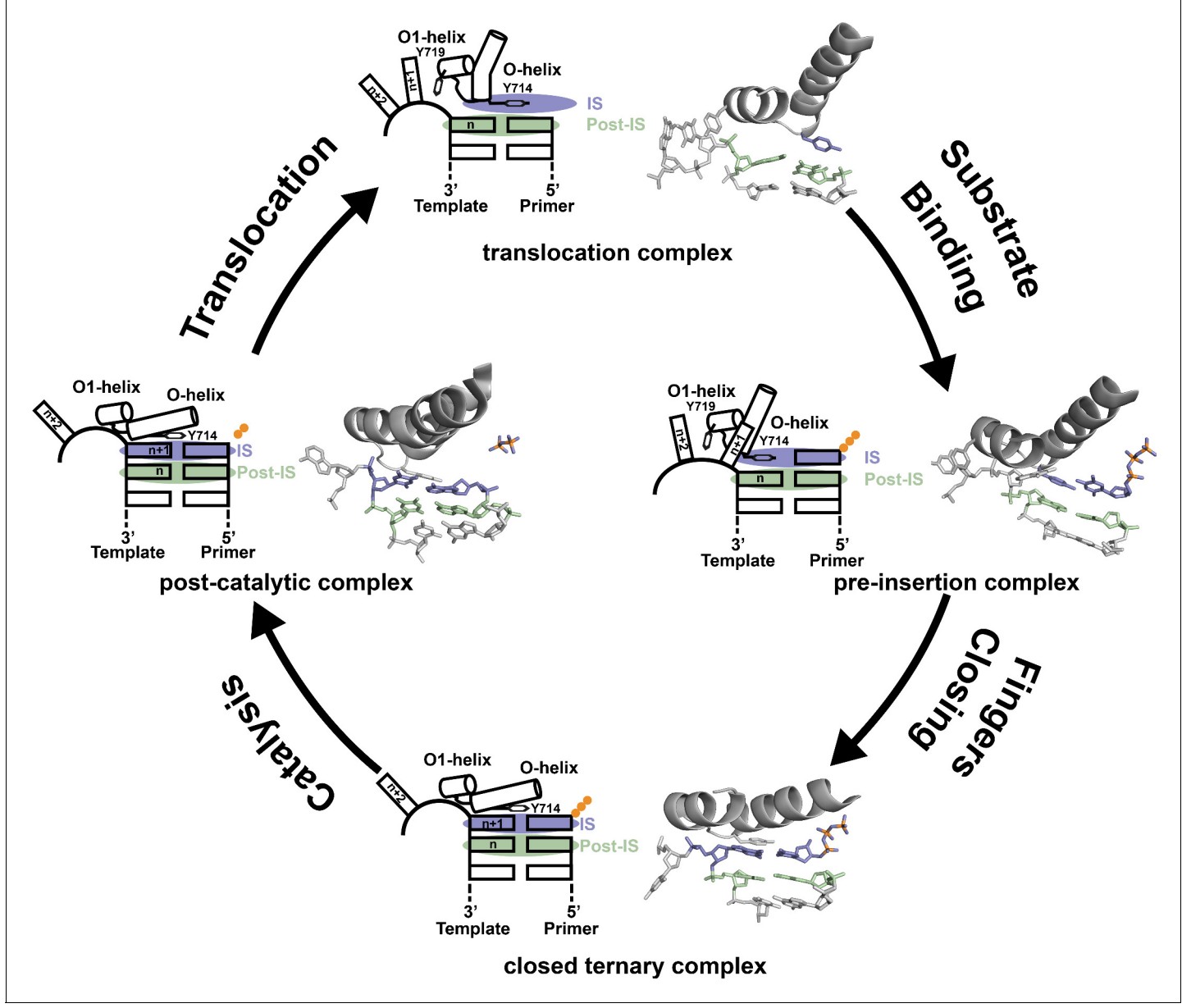

**Figure 3.** Revised mechanism of DNAP-I. The four key mechanistic steps of DNAP-I depict a replication cycle for DNA synthesis. The translocation complex (top) is stabilized by $\pi$-stacking interactions between Tyr[719] and the n + 1 templating base and between Tyr[714] and the primer strand. Tyr[714] occupies the insertion site (IS, purple) while a newly formed base pair is located in the post insertion site (post-IS, green). In the pre-insertion complex (right), the O-helix adjusts to accommodate the incoming dNTP substrate, which binds opposite Tyr[714] in the IS. In the closed ternary complex (bottom), the polymerase undergoes a major conformational change to allow the n + 1 templating base to form a nascent base pair with the dNTP substrate in pre-catalytic state. Following catalysis, the finger subdomain remains closed with a trapped pyrophosphate moiety observed in the active site of the post-catalytic complex (left). To complete the cycle, the finger subdomain opens, pyrophosphate is released, and the enzyme translocates to the next position on the template. The translocation (6DSY), pre-insertion (6DSU), and closed ternary complexes (1VL5) are based on crystal structures Bst DNAP-I. The post-catalytic complex is based on the structure of T7 RNAP (1S77), which is a homolog of Bst DNAP-I.

DOI: https://doi.org/10.7554/eLife.40444.011

## Bst cloning, Expression, and Purification

The *Bst* (amino acid residues 299–876) gene was PCR amplified from a previously constructed pDEST007-*Bst* vector generously donated by Prof Thomas Carell using Bst_for (ATC*CATATG*GCA TTTACGCTTGCTGAC, IDT) and Bst_rev (ATGCGGC*GGTCTC*C TCGAGTCATTATTTCGCATCATAC-CACG, IDT) primers containing *NdeI* and *BsaI* restriction enzyme sites (underlined), respectively. Purified PCR product and the expression vector, pGDR11, were digested with *NdeI* and *BsaI* restriction enzymes (NEB) and ligated and the resulting pGDR11-*Bst* construct was sequence verified (Retrogen). DH5-$\alpha$ cells (NEB) harboring pGDR11-*Bst* were grown aerobically at 37°C in LB medium containing 100 µg mL$^{-1}$ ampicillin. At an OD$_{600}$ of 0.8, expression of a tagless Bst was induced with 1 mM isopropyl β-D-thiogalactoside at 18°C for 16 hr. Cells were harvested by centrifugation for 20 min at 3315 x g at 4°C and lysed in 40 mL lysis buffer (50 mM Tris-Cl pH 7.5, 1 mM EDTA, 10 mM BME, 0.1 % v/v NP-40, 0.1 % v/v Tween20, 5 mg egg hen lysozyme) by sonication. The cell lysate was centrifuged at 23,708 x g for 30 min and the clarified supernatant was heat treated for 20 min at 60°C and centrifuged again at 23,708 x g for 30 min. The supernatant was loaded onto two 5 mL HiTrap Q HP columns (GE) assembled in tandem and washed with low salt buffer (50 mM Tris-Cl pH 7.5, 100 mM NaCl, 1 mM EDTA, 10 mM BME). Bst was eluted with a high salt buffer (50 mM Tris-Cl pH 7.5, 1M NaCl, 0.1 mM EDTA, 10 mM BME) using a linear gradient. Eluted fractions containing Bst were visualized by SDS-PAGE, pooled, and dialyzed against low salt buffer. The dialyzed sample was loaded onto a 5 mL HiTrap Heparin column (GE), washed with low salt buffer, and eluted using a linear gradient of high salt buffer. Eluted fractions containing Bst were visualized using SDS-PAGE and concentrated using a 30 kDa cutoff Amicon centrifugal filter (Millipore). Further purification was achieved by size exclusion chromatography (Superdex 200 HiLoad 16/600, GE) pre-equilibrated with Bst buffer (50 mM Tris-Cl pH 7.5, 150 mM NaCl, 1 mM EDTA, 10 mM BME). Purified Bst was concentrated to 20 mg mL$^{-1}$ for crystallization trials using a 30 kDa cutoff Amicon centrifugal filter (Millipore).

## Crystallization procedures

### General information

All reagents purchased from commercial suppliers were of analytical grade. Stock solutions of 2-methyl-2,4-pentanediol (Hampton Research), ammonium sulfate (Teknova) and 2-(*N*-morpholino) ethanesulfonic acid (Calbiochem) were filtered before use.

### Sample preparation

The DNA template (5'-GACGTACGTGATCGCA-3', T) and primer (5'-GCGATCACGT-3', P) strands, purchased from IDT, were used without further purification for crystallization trials. The P/T duplex (0.18 mM final concentration) was prepared by combining equal amounts of the primer and template strands in Bst buffer supplemented with 20 mM MgCl$_2$, and annealing the strands by heating at 95°C for 5 min and cooling to 10°C over 10 min.

### Crystallization

All polymerase samples were prepared at a final protein concentration of 4 mg mL$^{-1}$. The binary complex (n) was prepared by incubating Bst polymerase with three molar equivalents of the P/T duplex at 37°C for 30 min. For the primer extension complexes, the n sample was further incubated a second time with 10 M excess of dTTP (n + 1 complex) or dTTP +dATP (n + 2 complex) and 10 mM manganese chloride at 37°C for 30 min. Following primer-extension, 24–well plate hanging drop trays were used to monitor crystal growth over a range of ammonium sulfate and MPD concentrations, based on previously published conditions (*Johnson et al., 2003*). Each drop contained 1 µL of sample mixed with 1 µL of mother liquor over 500 µL of mother liquor per well. Trays were stored in the dark at room temperature and crystal growth was generally observed after 2 days. For the *in crystallo* extension, single crystals of the n + 1 extension product obtained from a solution-catalyzed reaction were transferred to a drop containing stabilization buffer (0.1 M MES pH 5.8, 2 M ammonium sulfate, 2.5% MPD) supplemented with 30 mM dATP and soaked for 4 days prior to harvesting. For the pre-insertion complex, single crystals of the n + 1 extension product obtained from a solution-catalyzed reaction were transferred to a drop containing stabilization buffer supplemented with

30 mM adenosine-5′-[(β,γ)-imido] triphosphate (dAMPNPP) and soaked for 5–6 days before harvesting.

## Data collection, structure determination, and refinement

Five diffraction datasets corresponding to n, n + 1, n + 1 dATP soak, n + 1 dAMPNPP soak, and n + 2 were collected at the Advanced Light Source (Lawrence Berkeley National laboratory, Berkeley, CA) from single crystals. Images were indexed, integrated, and merged using XDS (*Kabsch, 2010*). Data collection statistics are summarized in *Table 1*. Molecular replacement (MR) using Phaser (*McCoy et al., 2007*) was performed using PDB structures 1L3S, 1L3T, and 1L3U (*Johnson et al., 2003*) as search models for n, n + 1, and n + 1 dATP soak datasets, respectively. MR for dAMPNPP was performed using 1L3T (*Johnson et al., 2003*) as the search model and MR for n + 2 was performed using the n + 1 structure as the search model. All final models were determined using iterative rounds of manual building through Coot (*Emsley et al., 2010*) and refinement with phenix (*Afonine et al., 2012*). The final stages of refinement employed TLS parameters for all structures. The stereochemistry and geometry of all structures were validated with Molprobity (*Chen et al., 2010*), with the final refinement parameters summarized in *Table 1*. Final coordinates and structure factors have been deposited in the Protein Data Bank. All molecular graphics were prepared with PyMOL (*Delano, 2002*).

## Acknowledgements

We would like to thank T Poulos, A Luptak, and members of the Chaput laboratory for helpful discussions and critical reading of the manuscript. This work was supported by the DARPA Folded Non-Natural Polymers with Biological Function Fold F(x) Program under award number N66001-16-2-4061 and the National Science Foundation (MCB: 1607111). LJ was supported by undergraduate training grants from NIGMS (R25GM055246 and T34GM069337). Data sets were collected at the Advanced Light Source (ALS), which is supported by the DOE (Contract No. DE-AC02-05CH11231).

## Additional information

### Funding

| Funder | Grant reference number | Author |
| --- | --- | --- |
| Defense Advanced Research Projects Agency | N66001-16-2-4061 | John C Chaput |
| National Science Foundation | 1607111 | John C Chaput |
| National Institutes of Health | R25GM055246 | Lynnette N Jackson |

The funders had no role in study design, data collection and interpretation, or the decision to submit the work for publication.

### Author contributions

Nicholas Chim, Conceptualization, Data curation, Formal analysis, Funding acquisition, Investigation, Methodology, Writing—original draft, Project administration, Writing—review and editing, Conceived of the project and designed the experiments, Performed all the experiments that led to Bst DNAP-I crystals, Collected and processed all X-ray diffraction data sets; Lynnette N Jackson, Conceptualization, Data curation, Formal analysis, Investigation, Methodology, Writing—original draft, Project administration, Writing—review and editing, Performed all the experiments that led to Bst DNAP-I crystals, Collected and processed all X-ray diffraction data sets; Anh M Trinh, Data curation, Formal analysis, Investigation, Methodology, Writing—original draft, Writing—review and editing, Performed all the experiments that led to Bst DNAP-I crystals; John C Chaput, Conceptualization, Data curation, Formal analysis, Funding acquisition, Investigation, Writing—original draft, Project administration, Writing—review and editing, Conceived of the project and designed the experiments

**Author ORCIDs**

Nicholas Chim https://orcid.org/0000-0003-2274-5305
John C Chaput http://orcid.org/0000-0003-1393-135X

**Decision letter and Author response**
Decision letter https://doi.org/10.7554/eLife.40444.025
Author response https://doi.org/10.7554/eLife.40444.026

## Additional files

### Supplementary files

• Supplementary File 1. a.Helical statistics for solution and *in crystallo* catalyzed n + 1 translocated structures. b.Structures of Bst DNAP-I with homologous active sites.
DOI: https://doi.org/10.7554/eLife.40444.012

• Transparent reporting form
DOI: https://doi.org/10.7554/eLife.40444.013

### Data availability

Coordinates and structure factors have been deposited in the PDB with the accession codes: 6DSU, 6DSV, 6DSW, 6DSX, and 6DSY.

The following datasets were generated:

| Author(s) | Year | Dataset title | Dataset URL | Database and Identifier |
|---|---|---|---|---|
| Chim N, Jackson LN, Chaput JC | 2018 | Bst DNA polymerase I post-chemistry (n+1) structure | http://www.rcsb.org/structure/6DSY | RCSB Protein Data Bank, 6DSY |
| Chim N, Jackson LN, Chaput JC | 2018 | Bst DNA polymerase I pre-insertion complex structure | http://www.rcsb.org/structure/6DSU | RCSB Protein Data Bank, 6DSU |
| Chim N, Jackson LN, Chaput JC | 2018 | Bst DNA polymerase I post-chemistry (n+2) structure | http://www.rcsb.org/structure/6DSV | RCSB Protein Data Bank, 6DSV |
| Chim N, Jackson LN, Chaput JC | 2018 | Bst DNA polymerase I pre-chemistry (n) structure | http://www.rcsb.org/structure/6DSW | RCSB Protein Data Bank, 6DSW |
| Chim N, Jackson LN, Chaput JC | 2018 | Bst DNA polymerase I post-chemistry (n+1 with dATP soak) structure | http://www.rcsb.org/structure/6DSX | RCSB Protein Data Bank, 6DSX |

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
