## [Decision Letter]

Thank you for submitting your article "Crystal Structures of DNA Polymerase I Capture the Physiological Intermediates of Translocation and Pre-Insertion" for consideration by *eLife*. Your article has been reviewed by two peer reviewers, and the evaluation has been overseen by John Kuriyan as the Senior and Reviewing Editor. The reviewers have opted to remain anonymous.

The reviewers have discussed the reviews with one another and the Reviewing Editor has drafted this decision to help you prepare a revised submission.

This manuscript from Chim et al. describes the results of a crystallographic characterization of the Bst DNA polymerase I large fragment, a model enzyme for understanding the A-family DNA polymerases. The authors describe novel structures, compared with those in previous publications in the field and deposited in the PDB. Instead of relying on use of an in crystallo reaction set up toward following the reaction cycle, the authors used "direct" crystallization of complexes formed in solution. The crystal structures obtained in this work are generally similar to the crystal structures published by the group of Lorena Beese (Johnson et al., 2003; Kiefer et al., 1998), using the identical DNA polymerase and DNA substrates. In the original work, the different structures were obtained by extending a DNA substrate by one or two nucleotides in crystals. In the current work, the authors reproduce identical structures by performing the same in-crystal extension of the DNA substrate. The authors also present two new structures that are the results of performing the extension in solution before crystallization. These new structures differ from the in-crystal structures in that the template base that is to be paired with the incoming nucleotide (n+1 base) is not bound to the so-called pre-insertion site but positioned on the other side of a helix that borders the pre-insertion site.

In summary, the authors present structures of reaction intermediates in the DNA polymerase cycle that have not been observed previously. The results emphasize the point of dynamics in the fingers domain as the enzyme transitions from one step to the next in the reaction cycle.

We agree that this work is important, and potentially suitable for publication in *eLife*, because of the fundamental importance of DNA replication mechanisms.

We have serious concerns, however, with the way that the overall import of the work is currently presented, in terms of its potential "physiological relevance". All structural analyses are results of artifacts of the solution or crystal conditions, and it is likely that conformations that are stable enough to be trapped in one way or the other represent intermediates or transitions on an underlying free-energy surface that are inadequately sampled by current experimental methods and our inability to mimic cellular conditions. In particular, the authors argue that their new structures are "physiologically relevant", while those obtained previously through in-crystal extension are not. The authors provide no evidence to support this statement.

Their first argument is that the "pre-insertion site is inconsistent with the fast rate of Bst DNAP-I (~200nt/s)" (Introduction). Yet the authors do not explain why this would be inconsistent. They also claim that the binding of the n+1 base in the pre-insertion site "has not been witnessed in polymerases with homologous active sites" (Introduction). However, one of the referenced structures is without DNA, one is an RNA polymerase with a DNA-RNA hybrid substrate, and two of these are from a different species (*Thermus aquaticus*), both of which did not include a primer extension step. In contrast, one paper (Golosov, 2010) describes a molecular dynamics analysis that is completely consistent with the binding of the n+1 base into the pre-insertion site.

Curiously, the conformation of the DNA template observed in their new structure has been observed before, but only in structures "that contain damaged DNA intermediates and active site mutations" (Results and Discussion section). This might even argue against the new structures representing the "physiologically relevant state", at least for normal base pairs.

It is possible that both sets of structures could represent intermediates in a complex DNA extension cycle that contains many different intermediate structures. We do not feel that it is necessary to diminish the importance of previous contributions to gain value for this new work, which we find interesting in its own right.

We are prepared to consider a revised manuscript that takes into account the points made below. It is stressed, however, that the editors and the reviewers require that the Title and the language of the paper be changed so that a dichotomy, unsupported by additional data, not be presented between "physiologically relevant" and "physiologically irrelevant". No new experiments are requested.

Essential Revisions:

As noted above, change the Title of the paper to not imply physiological relevance or irrelevance.

It seems that the authors have a strong bias against the original, in-crystal extended structures, a bias that is voiced throughout the paper without much clarification. Please remove or modify these speculations in light of the comments made above. Some of these are pointed out below:

"…the pre-insertion site may be a constraint of the crystal lattice and not a relevant intermediate in the DNA synthesis pathway." (Introduction)

"Recognizing that in crystallo and solution catalyzed enzymatic reactions can produce different structural results with different functional interpretations," (Introduction)

"This work provides a telltale illustration of the problems that can arise when soaking substrates into preformed crystals of enzymes that undergo significant conformational changes (Ehrmann et al., 2017). To avoid such problems, we suggest validating important soaking results with cocrystallization."

Introduction, “[…] the pre-insertion site is inconsistent with the fast rate of Bst DNAP-I (~200 nt/sc) […]”: the inconsistency asserted is not clear and is not explained. These comments undercut the manuscript and are not meaningful as written.

In summary, the manuscript should be revised to highlight the impact of the present work without repudiating the significance of previous work.

---

## [Author Response]

We agree that this work is important, and potentially suitable for publication in eLife, because of the fundamental importance of DNA replication mechanisms.We have serious concerns, however, with the way that the overall import of the work is currently presented, in terms of its potential "physiological relevance". All structural analyses are results of artifacts of the solution or crystal conditions, and it is likely that conformations that are stable enough to be trapped in one way or the other represent intermediates or transitions on an underlying free-energy surface that are inadequately sampled by current experimental methods and our inability to mimic cellular conditions. In particular, the authors argue that their new structures are "physiologically relevant", while those obtained previously through in-crystal extension are not. The authors provide no evidence to support this statement.

We agree with the reviewers that, without additional experimental evidence, the claim of physiological relevance or irrelevance by any set of crystal structures is premature and unjustified. In the revised manuscript, we have removed all discussion of physiological relevance and presented all structural intermediates as a continuum of a complex DNA synthesis pathway.

Their first argument is that the "pre-insertion site is inconsistent with the fast rate of Bst DNAP-I (~200nt/s)" (Introduction). Yet the authors do not explain why this would be inconsistent. They also claim that the binding of the n+1 base in the pre-insertion site "has not been witnessed in polymerases with homologous active sites" (Introduction). However, one of the referenced structures is without DNA, one is an RNA polymerase with a DNA-RNA hybrid substrate, and two of these are from a different species (Thermus aquaticus), both of which did not include a primer extension step. In contrast, one paper (Golosov, 2010) describes a molecular dynamics analysis that is completely consistent with the binding of the n+1 base into the pre-insertion site.

We removed the claim that the pre-insertion site is inconsistent with the fast rate of Bst DNAP-I. However, we chose to retain the statement that the pre-insertion site has not been witnessed in polymerases with homologous active sites, as this sentence provides justification for the current work. In addition to changes to the text, references were provided to support the claims.

Curiously, the conformation of the DNA template observed in their new structure has been observed before, but only in structures "that contain damaged DNA intermediates and active site mutations" (Results and Discussion section). This might even argue against the new structures representing the "physiologically relevant state", at least for normal base pairs.

We chose to keep this paragraph in the manuscript as it describes a key observation that provides support for Tyr^714^ as a possible regulatory checkpoint. We do not feel that this paragraph diminishes previous work, as all assertions of physiological relevance or irrelevance have been removed from the text.

It is possible that both sets of structures could represent intermediates in a complex DNA extension cycle that contains many different intermediate structures. We do not feel that it is necessary to diminish the importance of previous contributions to gain value for this new work, which we find interesting in its own right.

The entire manuscript was revised to state that all intermediates identified in the current study and previous work represent individual steps in a complex DNA extension pathway.

We are prepared to consider a revised manuscript that takes into account the points made below. It is stressed, however, that the editors and the reviewers require that the Title and the language of the paper be changed so that a dichotomy, unsupported by additional data, not be presented between "physiologically relevant" and "physiologically irrelevant". No new experiments are requested.Essential Revisions:As noted above, change the Title of the paper to not imply physiological relevance or irrelevance.

We have changed the Title and revised the Abstract to remove any discussion of physiological relevance or irrelevance.

It seems that the authors have a strong bias against the original, in-crystal extended structures, a bias that is voiced throughout the paper without much clarification. Please remove or modify these speculations in light of the comments made above. Some of these are pointed out below:"…the pre-insertion site may be a constraint of the crystal lattice and not a relevant intermediate in the DNA synthesis pathway." (Introduction)

This sentence was removed from the text and surrounding sentences were modified.

"Recognizing that in crystallo and solution catalyzed enzymatic reactions can produce different structural results with different functional interpretations," (Introduction).

In the absence of any discussion of physiological relevance, we feel this sentence remains true and does not diminish the impact of previous work. However, we have included the word ‘potentially’ in the sentence to soften the language.

"This work provides a telltale illustration of the problems that can arise when soaking substrates into preformed crystals of enzymes that undergo significant conformational changes (Ehrmann et al., 2017). To avoid such problems, we suggest validating important soaking results with cocrystallization."

These sentences were removed from the text and the summary paragraph was re-written to be consistent with the revised text.

Introduction, “[…] the pre-insertion site is inconsistent with the fast rate of Bst DNAP-I (~200 nt/sc) […]”: the inconsistency asserted is not clear and is not explained. These comments undercut the manuscript and are not meaningful as written.

These lines were removed from the manuscript.

In summary, the manuscript should be revised to highlight the impact of the present work without repudiating the significance of previous work.

We have carefully rewritten the entire manuscript to present all structural intermediates, new and known, as a continuum of a complex DNA synthesis pathway.